# Cerebrospinal Fluid from Healthy Pregnant Women Does Not Harbor a Detectable Microbial Community

Yongyong Kang,[a,c,k] Xinchao Ji,[b] Li Guo,[c,d,e] Han Xia,[c,f] Xiaofei Yang,[a,d,g] Zhen Xie,[h] Xiaodan Shi,[i] Rui Wu,[i] Dongyun Feng,[i] Chen Wang,[j] Min Chen,[j] Wenliang Zhang,[d] Hong Wei,[d] Yuanlin Guan,[f] Kai Ye,[a,c,d,e,k] Gang Zhao[h,i]

[a]Genome Institute, The First Affiliated Hospital of Xi'an Jiaotong University, Xi'an, China
[b]Department of Neurology, Xi'an No. 3 Hospital, The Affiliated Hospital of Northwest University, Xi'an, China
[c]School of Automation Science and Engineering, Faculty of Electronic and Information Engineering, Xi'an Jiaotong University, Xi'an, China
[d]MOE Key Laboratory for Intelligent Networks & Network Security, Faculty of Electronic and Information Engineering, Xi'an Jiaotong University, Xi'an, China
[e]School of Life Science and Technology, Xi'an Jiaotong University, Xi'an, China
[f]Hugobiotech Co., Ltd., Beijing, China
[g]School of Computer Science and Technology, Faculty of Electronic and Information Engineering, Xi'an Jiaotong University, Xi'an, China
[h]School of Medicine, Northwest University, Xi'an, China
[i]Department of Neurology, Xijing Hospital, The Fourth Military Medical University, Xi'an, China
[j]Department of Anesthesiology, Xijing Hospital, The Fourth Military Medical University, Xi'an, China
[k]Center for Mathematical Medical, The First Affiliated Hospital of Xi'an Jiaotong University, Xi'an, China

Yongyong Kang, Xinchao Ji, Li Guo, and Han Xia contributed equally to this article. Author order was determined by the author who carried the final responsibility for the manuscript.

**ABSTRACT** Cerebrospinal fluid (CSF) circulating in the human central nervous system has long been considered aseptic in healthy individuals, because normally, the blood-brain barrier can protect against microbial invasions. However, this dogma has been called into question by several reports that microbes were identified in human brains, raising the question of whether there is a microbial community in the CSF of healthy individuals without neurological diseases. Here, we collected CSF samples and other samples, including one-to-one matched oral and skin swab samples (positive controls), from 23 pregnant women aged between 23 and 40 years. Normal saline samples (negative controls), sterile swabs, and extraction buffer samples (contamination controls) were also collected. Twelve of the CSF specimens were also used to evaluate the physiological activities of detected microbes. Metagenomic and metatranscriptomic sequencing was performed in these 116 specimens. A total of 620 nonredundant microbes were detected, which were dominated by bacteria (74.6%) and viruses (24.2%), while in CSF samples, metagenomic sequencing found only 26 nonredundant microbes, including one eukaryote, four bacteria, and 21 viruses (mostly bacteriophages). The beta diversity of microbes compared between CSF metagenomic samples and other types of samples (except negative controls) was significantly different from that of the CSF self-comparison. In addition, there was no active or viable microbe in the matched metagenomic and metatranscriptomic sequencing of CSF specimens after subtracting those also found in normal saline, DNA extraction buffer, and skin swab specimens. In conclusion, our results showed no strong evidence of a colonized microbial community present in the CSF of healthy individuals.

**IMPORTANCE** The microbiome is prevalent throughout human bodies, with profound health implications. However, it remains unclear whether it is present and active in human CSF, which has been long considered aseptic due to the blood-brain barrier. Here, we applied unbiased metagenomic and metatranscriptomic sequencing to detect the presence of a microbiome in CSF collected from 23 pregnant women with matched controls. Analysis of 116 specimens found no strong evidence to support the presence of a colonized microbiome in CSF. Our findings will strengthen our understanding of the internal

Address correspondence to Kai Ye, kaiye@xjtu.edu.cn, or Gang Zhao, zhaogang@nwu.edu.cn.

environment of the CSF in healthy people, which has strong implications for human health, especially for neurological infections and disorders, and will help further disease diagnostics, prevention, and therapeutics in clinical settings.

**KEYWORDS** cerebrospinal fluid, next-generation sequencing, metagenomics, pathogen, microbiome

First defined by Joshua Lederberg in 2001 (1), the human microbiome has since been discovered in almost every part of the human body, including the gut, oral cavity, skin, bladder, vagina, and lungs (2–8). It has profound impacts on human health, including being associated with a broad range of human diseases, including cancers, diabetes, schizophrenia, and autoimmune diseases (9–12). However, due to the difficulties in the identification and traceability of contamination, it remains controversial whether there is a colonized microbial community at some body sites, such as the placenta, blood, and amniotic fluid (13–17).

Cerebrospinal fluid (CSF), circulating in the human central nervous system (CNS), has long been considered sterile, given that the blood-brain barrier can effectively protect against microbial invasions. However, the dogma has been challenged in recent years by several reports of microbes being detected in human brains and CSF. For example, the bacterial pathogen *Porphyromonas gingivalis* was identified in brain regions, including the cerebral cortex and hippocampus, in patients with Alzheimer's disease (18). In addition, a number of DNA viruses were identified in the CSF of mostly healthy individuals (19). Although both experimental and analytical methods have recently improved in sensitivity and accuracy, it remains elusive whether these reports are evidence of the existence of a common microbiome in human CSF and CNS or simply sporadic and accidental events.

Given the debate over the existence of any microbial community in CSF and the importance of understanding microbial infection in the human CNS, we have performed microbiome analysis to characterize bacteria, archaea, eukaryotes, and viruses in CSF samples from 23 donors without neurological disorders, as well as one-to-one matched positive controls (oral and skin swab samples) and negative controls (normal saline). DNA/RNA extraction buffers and sterile swabs were collected as contamination controls. In total, 116 specimens were used in this study. Considering the limitations of the 16S rRNA gene-based approach in identifying microbes to the species or strain level (20–23), unbiased metagenomic next-generation sequencing (mNGS) and metatranscriptomic next-generation sequencing methods were chosen for detecting the total DNA and RNA of microbes at a species resolution and assessing the physiological status of microbes detected in CSF samples (24, 25). As a promising approach, the clinical diagnostic performance of mNGS for infectious diseases has been widely adopted in the medical community, following multicenter studies (26–28).

## RESULTS

**mNGS results of all specimens.** A total of 116 specimens were collected, including 23 CSF specimens for metagenomic sequencing (CSF DNA) and 12 CSF specimens for metatranscriptomic sequencing (CSF RNA), 23 normal saline specimens as negative controls for CSF sampling, 23 skin and 23 oral swab specimens as positive controls, and 6 sterile swabs and 6 DNA/RNA extraction buffer specimens as contamination controls (Fig. 1A and Fig. S1 in the supplemental material). In total, we detected 620 nonredundant microbes from the 116 specimens that were identified to the species level using metagenomic and metatranscriptomic sequencing (Table S1). These microbes were dominated by bacteria (74.6%) and viruses (24.2%). Overall, skin and oral swab samples and sterile swabs had the most abundant microbiomes of all samples with 393, 199, and 137 nonredundant microbes, respectively. In contrast, the number of nonredundant microbes detected in CSF DNA ($n = 26$), negative controls ($n = 49$), and extraction buffers ($n = 27$) were relatively fewer (Fig. 1B). The swab samples, including skin and oral swab samples and even sterile swabs, contained various numbers of microbial DNAs (Fig. 1B and Fig. S2). One reason was that sterilization experiments could

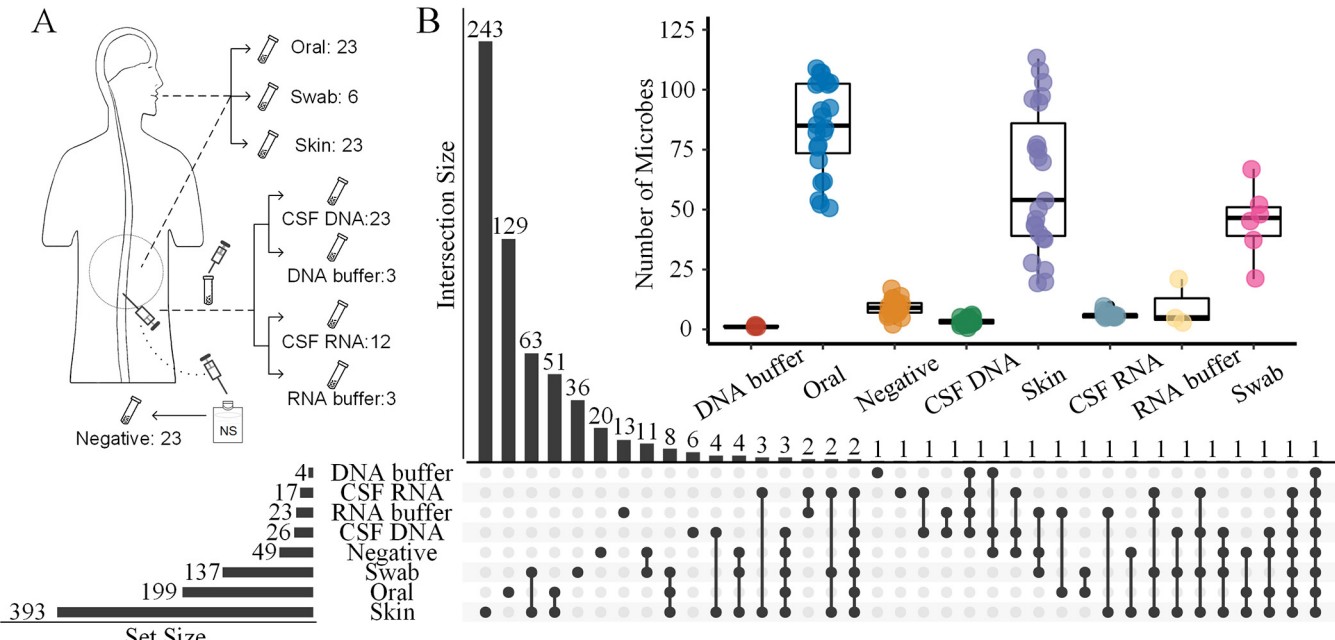

**FIG 1** Study design and numbers of microbes detected in different of types of specimens. (A) Experimental design in this study. CSF and matched control samples (positive controls, oral and skin swab samples; negative controls, saline solution) collected from 23 pregnant women, along with DNA/RNA extraction buffer samples (number indicates the number of specimens) and sterile swabs were sequenced for metagenomic and metatranscriptomic analysis (see Materials and Methods). (B) An overview of microbes detected in each sample type. The number of microbes detected in each sample and species shared between different samples are shown in the UpSet plot, with the dots representing intersections among sample types and the bars representing the number of microbes for each sample type (horizontal bars) and the number shared for each intersection type (vertical bars). The inset shows a box plot summarizing the distributions of the numbers of species detected for different sample types.

only kill the microbes, while various microbial DNAs remained on sterile swabs, which could be detected by mNGS due to its high sensitivity. We then removed the species detected in sterile swab samples (contamination controls) from those detected in the skin and oral swab samples, after which 78% and 92% of species detected still remained in the skin and oral swab samples, respectively. This came as no surprise, because the skin and the oral cavity are well known to harbor a plethora of microbes.

The species detected in different types of specimens were also compared, finding little overlap among all samples. Skin and oral swab samples and sterile swabs had large numbers of unique microbes among all sample types, with 243, 129, and 36 taxa, respectively, found only in these samples (Fig. 1B). For each sample type, the numbers of microbes ranged widely in skin and oral swab samples and sterile swabs but more narrowly in CSF DNA samples, CSF RNA samples, negative controls (normal saline), and DNA/RNA extraction buffer samples (Fig. 1B). The oral swab samples were rich in *Streptococcus*, *Veillonella*, *Neisseria*, *Rothia*, and *Prevotella*, while the skin swab samples were rich in *Cutibacterium*, *Staphylococcus*, *Micrococcus*, and *Malassezia* (Fig. S3). These results were consistent with previous studies (4, 6, 29), which provided a proof-of-concept of the NGS-based metagenomic sequencing method, laying a solid foundation for our exploration of the CSF microbiome using such a method.

In CSF DNA specimens, a total of 26 nonredundant microbes, including four bacteria, 21 viruses, and one eukaryote taxon were detected, and up to 6 microbes were found in a single specimen (Fig. 2A). Most of the viruses were bacteriophages. The relative abundances of microbes suggested that the species cyprinid herpesvirus 3 was the predominant species in 19 of 23 CSF DNA specimens (Fig. 2B). Additionally, 100%, 26%, 22%, and 22% of all CSF DNA specimens contained cyprinid herpesvirus 3, human alphaherpesvirus 2, Enterobacteria phage mEp460, and dasheen mosaic virus, respectively. However, cyprinid herpesvirus 3, detected in all CSF DNA specimens, was also found in all negative controls and skin swab specimens, suggesting a likely external source of this microbe during the CSF sampling procedure. Although *Aspergillus*

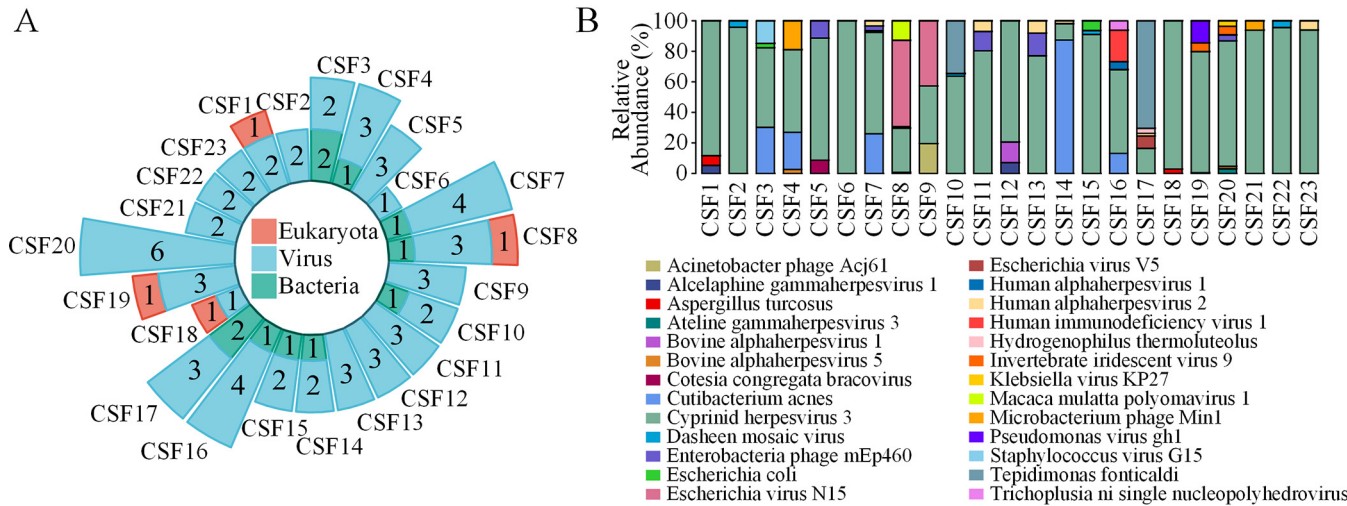

**FIG 2** Microbial community structure in CSF of 23 healthy individuals. (A) Circle bar plot summarizing the number(s) of microbial species in each CSF DNA specimen, categorized into three major types: eukaryotes, viruses, and bacteria. (B) Microbial community structures of 23 CSF DNA specimens shown in a stacked bar plot that summarizes the relative abundances of the different species of microbes detected in each CSF DNA specimen.

*turcosus*, a eukaryote, appeared in four specimens, its relative abundances were very low, at 0.45%, 0.66%, 2.87%, and 6.38%. The bacteria *Cutibacterium acnes* and *Tepidimonas fonticaldi* were the predominant species in 2 CSF DNA specimens (Fig. 2B).

**The microbiome signatures of cerebrospinal fluid and negative controls were similar.** To rule out microbes that might have been introduced into CSF specimens during the sampling process and mNGS experiments, the mNGS results of CSF and other types of specimens (skin and oral swab samples, sterile swabs, and normal saline) were compared in this study. Nonmetric multidimensional scaling (NMDS) analysis (Fig. 3), principal coordinate analysis (PCoA) (Fig. S4), and beta diversity analysis revealed an overall clear separation of microbial communities between CSF DNA and other types of specimens, except negative controls (Fig. 4). In addition, the diversities of the microbial communities detected in CSF DNA and negative controls were low. There was no significant difference in the beta diversities for the CSF DNA-versus-negative control comparison and the CSF DNA self-comparison (Wilcoxon test, $P = 0.59$) (Fig. 4), but the beta diversity results between CSF DNA and the other specimen types were significantly different from the CSF DNA self-comparison. In fact, the microbes shared between CSF DNA specimens and negative controls accounted for 42% and 22% of all microbes detected in CSF DNA specimens and negative controls, respectively. In addition, 58% of microbes from CSF DNA were also found in skin swab speci-

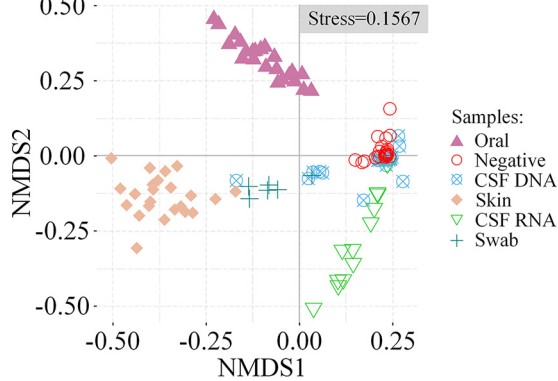

**FIG 3** Microbiome similarities among specimen types. NMDS (nonmetric multidimensional scaling) analysis of microbial species detected from different specimen types. Shapes and colors represent specimen types as shown in the key.

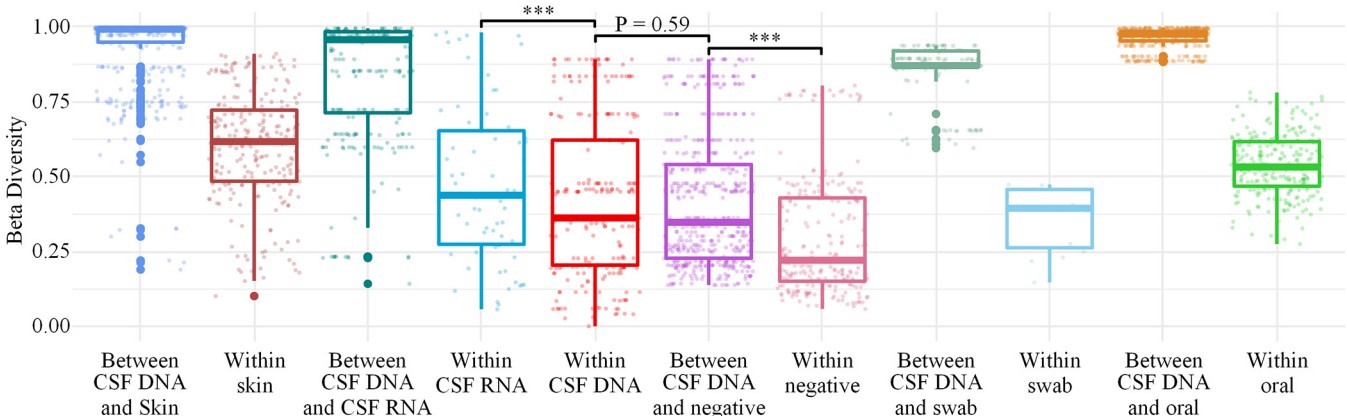

**FIG 4** The bacterial microbiota of CSF DNA are indistinguishable from that of negative controls. Box plot summarizing the beta diversities within CSF DNA specimens and between CSF DNA and other specimens (skin swab samples, oral swab samples, sterile swabs, and negative [normal saline]) using Bray-Curtis dissimilarity. Statistical significance was assessed by Wilcoxon test, whose significance level is indicated with asterisks (***, $P < 0.001$).

mens. These results indicate that the DNA detected in CSF samples may have come partly from negative controls or skin during sample collection.

**No microbiome was present in the CSF after subtracting microbes found in controls.** We questioned whether these detected microbes were truly CSF inhabitants or simply brought in from external sources, such as skin, sampling equipment, and DNA extraction buffer. To reduce the exogenous DNA noise of CSF samples, we subtracted the microbes collectively detected in negative controls (normal saline) and DNA extraction buffer specimens from the microbes of each CSF DNA specimen. After subtraction, 12 CSF DNA specimens had no remaining microbes, whereas the other 11 CSF DNA specimens contained a total of 14 microbes, including 11 viruses, 2 bacteria, and 1 eukaryote (Fig. 5). Since an introduction of microbes from skin could not be completely ruled out, we further subtracted the microbes from skin swab specimens. A total of 8 microbes in 9 CSF DNA specimens were found. The remaining 8 microbes, which were potentially CSF-inhabiting microbes, included five viruses (bovine alphaherpesvirus 1, *Escherichia* virus V5, *Klebsiella* virus KP27, *Macaca mulatta* polyomavirus 1, and Trichoplusia ni single nucleopolyhedrovirus), two bacteria (*Hydrogenophilus thermoluteolus* and *Tepidimonas fonticaldi*), and one eukaryote (*Aspergillus turcosus*).

The detection of microbes using metagenomic sequencing offers a glimpse of microorganisms present in certain niches. However, it remains uncertain whether these microbes

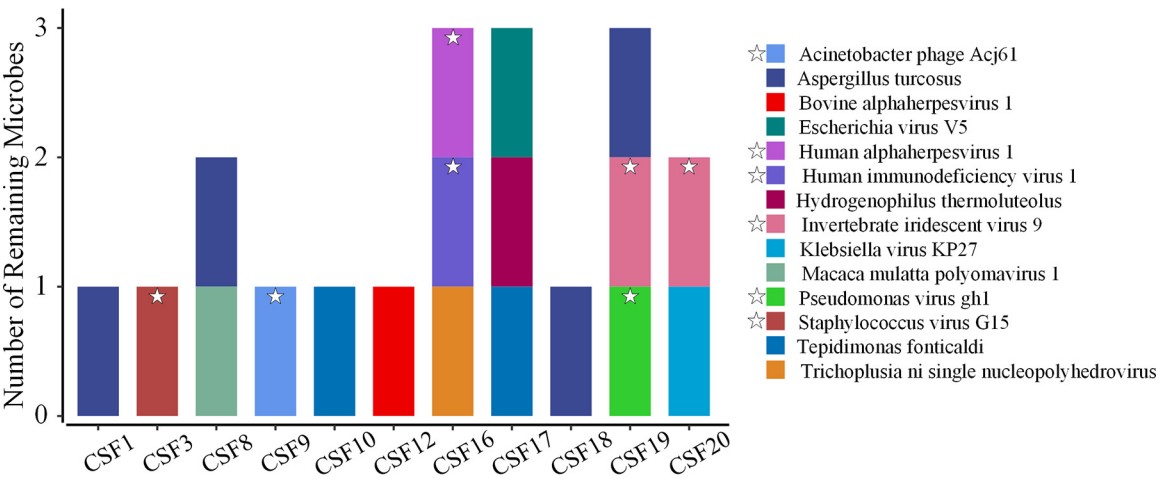

**FIG 5** Microbes remaining in the cerebrospinal fluid after subtracting the microbes that appeared in the negative controls (normal saline) and DNA extraction buffer. Fourteen species (the 6 species labeled with stars also appeared in skin swab samples) remained in CSF DNA specimens.

are alive or dead, as DNA from dead cells can also be detectable by mNGS. To further confirm whether the microbes detected in CSF specimens were alive or just DNA fragments from dead microbes from other sites of the body, we evaluated the physiological activities of the microbes potentially inhabiting CSF specimens using metatranscriptomic sequencing. Microbes detected by both the metagenomic and metatranscriptomic approach would be indicated to be active in the CSF. CSF transcriptomics revealed transcripts of human endogenous retrovirus K, equine infectious anemia virus, dasheen mosaic virus, and cyprinid herpesvirus 3 in all CSF RNA specimens and *Escherichia coli* in 11 specimens. Interestingly, when only considering the specimens with suspected colonized microbes by metagenomic sequencing after subtraction of microbes found in negative controls and DNA extraction buffers, bovine alphaherpesvirus 1 and human alphaherpesvirus 1 were found in CSF by both metagenomics and metatranscriptomics. However, human alphaherpesvirus 1 was also detected in skin swab specimens and bovine alphaherpesvirus 1 appeared in unmatched CSF DNA and CSF RNA specimens. It is worth noting that we detected bovine alphaherpesvirus 5 in 13 negative controls and 1 skin swab specimen. At the genus level, simplexvirus and varicellovirus, which appeared in both CSF DNA and CSF RNA specimens, also appeared in negative controls and skin swab specimens, suggesting that these species potentially originated from skin or other contaminations. Although metagenomic analysis detected the *Aspergillus turcosus* species in specimens from four individuals (Fig. 5), no transcripts of *Aspergillus turcosus* were detected in metatranscriptomic sequencing, suggesting a lack of living cell activity. Our study found no strong evidence of a colonized microbial community in the CSF of healthy individuals.

## DISCUSSION

In this study, CSF specimens were collected from a cohort of 23 healthy individuals without neurological disease, accompanied by a matched set of controls. A culture-independent approach was used to detect total DNA and RNA of microbes. The possibility of a microbiome existing in the CSF of healthy people was evaluated. As positive controls, skin and oral swab samples had more abundant microbiomes and higher beta diversities than CSF specimens. The blood-brain barrier could protect against invasions of most microbes into the CSF, especially for healthy individuals. However, we also found no significant difference in beta diversities between the CSF DNA-versus-negative control comparison and the CSF DNA self-comparison. NMDS and PCoA analysis also showed similar compositions of microbes among CSF DNA specimens and negative controls. This strengthened the possibility that the microbes detected in CSF specimens were from exogenous contamination.

We then removed the microbes from the CSF DNA specimen data that were also found in negative controls (normal saline), contamination controls (DNA extraction buffer), and positive controls (skin swab specimens). After the subtraction, 8 microbes that were potentially CSF-inhabiting microbes were obtained. Viruses were the most common before and after subtraction. Interestingly, most of these viruses were bacteriophages. Previous studies also found bacteriophages in CSF specimens (19). However, most bacteriophages obtained in CSF specimens were also detected in skin and normal saline specimens in this study. Thus, there was no clear evidence of colonization by bacteriophages in the CSF of healthy individuals.

Four individuals had *Aspergillus turcosus* detected in CSF DNA specimens. *Aspergillus turcosus* is well known as an opportunistic pathogen and can infect individuals with compromised immune systems. However, it is rarely reported in clinical samples (30, 31). In addition, no transcripts of *Aspergillus turcosus* were detected in CSF RNA specimens; therefore, the detected *Aspergillus turcosus* more likely represented DNA fragments than microbial cells with limited activity. In fact, no microbe identified at the genus level was detected by both metagenomic and metatranscriptomic sequencing after subtracting its occurrence in controls.

We focused on determining whether a CSF microbiome is present in healthy individuals without neurological disorders, a long-disputed issue in scientific and clinical research fields. Our findings demonstrated no strong evidence of a colonized microbial community in the

CSF of healthy individuals. In addition, although the possibility is intriguing, it remains unclear whether a microbiome is present in the CSF of patients diagnosed with diseases like Alzheimer's disease, multiple sclerosis, and Parkinson's disease and what roles a CSF microbiome possibly plays in the development of these disorders.

mNGS is a powerful tool for detecting microbiomes at a species resolution, especially for microbiome studies in specimens of low-abundance biomass, such as CSF. It has become an important auxiliary method for clinical pathogenic diagnosis and treatment of infectious diseases. mNGS has a high sensitivity and can detect all DNA fragments, not only from live microbial cells but also DNA fragments released from dead microbes in other sites (such as peripheral blood and tissues) of the body, experiment reagents, and consumables. In addition, during the lumbar puncture, tissues such as skin, muscle, and blood vessels can also be potential sources of contamination. However, due to technical and ethical restrictions in the actual operation process, it is difficult to collect specimens of all kinds of tissues that are exposed during the sampling process. For example, it is very difficult to collect muscle and blood specimens during the collection of CSF, especially for healthy people. Moreover, we need to be very cautious when designing a control, because whether the control itself contains microbes will also be controversial. Thus, strict disinfection measures before the operation and the construction of a database of colonizing microorganisms of these tissues will help filter out noise signals and reduce the false-positive rate.

The main challenge of this study is an overall lack of CSF samples from healthy human subjects. Only 23 pregnant females aged from 23 to 40 were enrolled. This is because lumbar puncture to collect CSF is an invasive surgery and will cause pain and damage to the subjects. Usually, only patients with CNS infections will receive lumbar punctures. Lumbar puncture in healthy people is rare. In addition, technically sound sampling, as well as data analysis methods based on different reference databases and taxonomic strategies, can also result in some biases.

In conclusion, using metagenomic combined with metatranscriptomic deep sequencing, we found that the microbiome profile in CSF specimens was indistinguishable from that in negative controls, indicating no strong evidence of a colonized microbial community in the CSF of healthy individuals. Such findings will strengthen our understanding of the internal environment of the CSF in healthy people, which has strong implications for human health, especially for neurological disorders and infections, and will help to further disease diagnostics, prevention, and therapeutics in clinical settings.

## MATERIALS AND METHODS

**Subjects.** A total of 23 donors who needed intraspinal anesthesia before caesarean section were recruited from the Department of Obstetrics, Xijing Hospital of the Fourth Military Medical University, during the summer of 2018. Subjects who had suffered from central nervous system infectious diseases (e.g., meningitis and encephalitis), systemic infectious diseases (hepatitis and tuberculosis), or autoimmune diseases (e.g., systemic lupus erythematosus and rheumatism) or had received antibiotic treatment in the past 6 months were excluded. Subjects with a history of hypertension, diabetes, heart disease, cancer, or neurological diseases (such as Alzheimer's disease, Parkinson's disease, multiple sclerosis, and epilepsy) were also excluded. This study was approved by the Ethics Committee of the Xijing Hospital of the Fourth Military Medical University. All procedures were conducted in accordance with the approved guidelines. All donors read and signed the consent form before sample collection.

**Sample collection.** To investigate whether there is a microbiome in CSF, we collected CSF samples from 23 pregnant women aged 23 to 40 years who underwent intraspinal anesthesia via lumbar puncture before caesarean section, and we coupled these samples with normal saline samples collected by syringe as negative controls. For each subject, oral and skin swab specimens were also collected as one-to-one-matched positive controls. Six sterile swabs and 3 DNA extraction buffer specimens were also collected as contamination controls. All samples were then subjected to DNA extraction and metagenomic sequencing. Finally, to validate whether the microbiome, if any, detected in CSF was physiologically active, metatranscriptomic sequencing was performed for 12 of the pregnant women's CSF specimens. Three RNA extraction buffer specimens were also collected as contamination controls.

Lumbar puncture was performed in the 23 subjects enrolled in this study. CSF was collected into a 4-ml centrifuge tube and then transferred to a $-80°C$ freezer for metagenomics sequencing. Twelve CSF specimens were randomly selected from the 23 pregnant women for metatranscriptomic studies (when processing different batches of CSF samples, we randomly selected samples for both metagenomic and metatranscriptomic sequencing, and used the remaining samples just for metagenomic sequencing), and RNA protection reagent was added to the CSF immediately after collection. The skin of the back of

each individual in the 5- by 5-cm$^2$ area around the puncture site (L3-L4 intervertebral space) was swabbed using a sterile cotton swab before the skin was cleaned with povidone iodine. To maximize the microbial load, no bathing was permitted within the 24 h before sample collection. For oral swab samples, the surfaces of the tongue, buccal fold, hard palate, soft palate, teeth, gingiva, and saliva (attached to the oral environment) were swabbed with sterile swabs. All subjects fasted for at least 6 h before the operation. Unused sterile swabs were also collected as contamination controls. Details of collected samples are described in Fig. S1 and Table S2.

**DNA extraction and purification.** DNA was extracted using the QIAamp DNA minikit (Qiagen, Germany) according to the manufacturer's instructions. Swab tips were cut into a 2-ml microcentrifuge tube. A total of 400 $\mu$l of phosphate-buffered saline (PBS), 20 $\mu$l of proteinase K, and 400 $\mu$l of buffer AL was added into the tube. After eddying for 10 s, swab samples were incubated at 56°C for 15 min, and 400 $\mu$l ethanol (100%) was mixed again with each sample. Samples were then transferred into QIAamp mini-spin columns. After washing with buffer AW1 and AW2, DNA of each sample was stored in 35 $\mu$l of buffer EB at −20°C. From the CSF and normal saline samples, 200 $\mu$l of each sample was mixed with 20 $\mu$l of proteinase K and 200 $\mu$l of buffer AL. After incubating at 56°C for 15 min, 200 $\mu$l ethanol (100%) was added and mixed in. DNA was then extracted as described above.

**DNA library construction.** The QIAseq FX DNA library kit (Qiagen, Germany) was used to construct DNA libraries. From each sample, 32.5 $\mu$l purified DNA was fragmented into 200- to 300-bp segments by incubation with 5 $\mu$l of FX buffer, 2.5 $\mu$l of FX enhancer, and 10 $\mu$l of FX enzyme mixture in cycles of 4°C for 1 min, 32°C for 12 min, and 65°C for 30 min. Five microliters of adaptor, 20 $\mu$l of ligation buffer, 10 $\mu$l of DNA ligase, and 15 $\mu$l of nuclease-free water were added, and the mixture incubated at 20°C for 15 min to initiate adapter ligation. Adapter ligation cleanup was performed immediately using 80 $\mu$l of resuspended AMPure XP beads (0.8×). After incubating at room temperature for 5 min, the beads were pelleted on a magnetic stand (Invitrogen) for 2 min. The supernatant was discarded, the pellet was washed twice with 200 $\mu$l of 80% ethanol, and then the beads were eluted with 52.5 $\mu$l of buffer EB. Subsequently, the supernatant was transferred into a new 1.5-ml microcentrifuge tube for a second purification using 50 $\mu$l AMPure XP beads (1×). The purified DNA was then used to construct the DNA library using the QIAseq FX DNA library kit (Qiagen, Germany). After mixing the library product with 25 $\mu$l of HiFi PCR master mix and 1.5 $\mu$l of primer mix, PCR enrichment was performed under cycling conditions of 2 min at 94°C, 12 cycles of 20 s at 98°C, 30 s at 60°C, and 30 s at 72°C, and 1 min at 72°C. The PCR products were finally purified with AMPure XP beads as described above.

**RNA extraction and purification.** Total RNA was extracted using the RNeasy minikit (Qiagen, Germany) according to the manufacturer's instructions. The pellet of each sample that had been treated with RNA protection reagent as described above was resuspended in 100 $\mu$l Tris-EDTA (TE) buffer containing lysozyme. Proteinase K was added, and the mixture incubated at room temperature for 10 min. After mixing in 350 $\mu$l of buffer RLT, RNA isolation and purification were performed with buffers AW1 and RPE, respectively, using the RNeasy mini-spin column, followed by elution with RNase-free water.

**RNA library preparation for metatranscriptomic sequencing.** The QIAseq FX single-cell RNA library kit (Qiagen, Germany) was used to construct RNA libraries. A total of 8 $\mu$l of purified RNA and 3 $\mu$l of NA denaturation buffer were added into a sterile PCR tube and incubated at 95°C for 3 min. To remove genomic DNA (gDNA), 2 $\mu$l of gDNA wipeout buffer was added and the mixture incubated at 42°C for 10 min. Four microliters of RT buffer, 1 $\mu$l of random primer, 1 $\mu$l of oligo(dT) primer and 1 $\mu$l of Quantiscript RT enzyme mix were added to each sample prior to RT at 42°C for 60 min. Eight microliters of ligase buffer and 2 $\mu$l of ligase mix were added into the reverse transcription reaction mixture and incubated at 24°C for 30 min. Then, 1 $\mu$l of REPLI-g SensiPhi DNA polymerase and 29 $\mu$l of reaction buffer were used for multiple displacement amplification (MDA) at 30°C for 2 h. A length of approximately 2,000 to 70,000 bp of amplified cDNA was finally produced. The amplified cDNA was diluted 1:3 in H$_2$O sc, and 10 $\mu$l of the diluted DNA and FX enzyme mix was used to obtain 300-bp library sequences with the following reaction conditions: 4°C for 1 min, 32°C for 15 min, 65°C for 30 min, and a 4°C hold. Five microliters of adapter and 45 $\mu$l of ligation master mix were added into each sample, and the mixture incubated at 20°C for 15 min. Subsequently, the adapter ligation cleanup was performed with AMPure XP beads as described above. The purified libraries were finally obtained for sequencing without further PCR amplification.

**Next-generation sequencing.** Shotgun sequencing was performed on the Illumina HiSeq platform for all samples (paired-end library with 150-bp read length). Approximately 25 Gb and 5 Gb of raw paired-end reads were obtained per sample in CSF DNA-positive and -negative samples, respectively.

**Data quality control.** To reduce the impact of host reads, we needed to remove human reads from the raw sequencing data before bioinformatics analysis. KneadData (version 0.7.4) (32), a widely used tool, is designed to perform quality control on metagenomic and metatranscriptomic sequencing data, especially for microbiome experiments. All reads were filtered using KneadData with the Trimmomatic options ILLUMINACLIP, TruSeq3-PE-2.fa:2:30:10:8:true, SLIDINGWINDOW, 4:20, and MINLEN, 50, and the bowtie2 options –very-sensitive and –dovetail. The proportions of human reads in CSF genomics samples were up to 92%.

**Detecting potential microbiome.** MetaPhlAn (version 3.0.1) (33) is a computational tool for profiling the composition of microbial communities (bacteria, archaea, viruses, and eukaryotes) from shotgun sequencing data. Based on ~1.1 million unique clade-specific marker genes identified from ~100,000 reference genomes, MetaPhlAn can profile unambiguous taxonomic assignments and accurate estimation of relative abundances of organisms in species-level resolution. Classifying the reads according to the marker genes database, MetaPhlAn outputs a file containing detected microbes and their relative abundances. MetaPhlAn was run with the custom parameters –add_viruses –input_type fastq –

read_min_len 50. It is worth noting that MetaPhlAn (version 2) was the only bioinformatics tool with 0% false-positive relative abundance and provided the best diversity estimate (34). In a case where one clade represented multiple species, only the representative species listed in column 1 was used. The numbers of redundant and nonredundant microbial taxa, respectively, represent the cumulative total number of species detected and the total number of different species detected in a certain specimen type.

**$\beta$-Diversity and phylogenetic analysis.** $\beta$-Diversity (between-sample diversity) was estimated by Bray-Curtis dissimilarity in the vegan package of R (version 4.1.0). All figures were also plotted using R.

**Availability of data and materials.** The clean sequence data reported in this paper have been deposited in the Genome Sequence Archive in the BIG Data Center (35, 36), Chinese Academy of Sciences, under accession number CRA004315, which is publicly accessible at https://ngdc.cncb.ac.cn/gsa.

## SUPPLEMENTAL MATERIAL

Supplemental material is available online only.
**SUPPLEMENTAL FILE 1**, PDF file, 0.5 MB.
**SUPPLEMENTAL FILE 2**, XLSX file, 0.2 MB.

## ACKNOWLEDGMENTS

We thank Peng Jia, Tingjie Wang, Ningxin Dang, Honghui Shen, and Tun Xu for helpful discussions regarding data analysis and Jing Hai for administrative and technical support. We thank the High-Performance Computing Cluster of the First Affiliated Hospital of Xi'an Jiaotong University for data processing.

This study was supported by the National Key R&D Program of China (grants no. 2018YFC0910400, 2017YFC0907500, and 2016 YFC0904501), the National Natural Science Foundation of China (grants no. 31671372, 61702406, 31701739, and 31970317), and the National Science and Technology Major Project of China (grant no. 2018ZX10302205), as well as general financial grants from the China Postdoctoral Science Foundation (grants no. 2017M623178 and 2017M623188).

We declare no conflicts of interest.

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
