## [Reviewer comments · Microbiology Spectrum]

Microbiology Spectrum

Cerebrospinal fluid from healthy pregnant women does not harbor a detectable microbial community

yongyong kang, Xinchao Ji, Li Guo, Han Xia, Xiaofei Yang, Zhen Xie, Xiaodan Shi, Rui Wu, Dongyun Feng, Chen Wang, Min Chen, Wenliang Zhang, Hong Wei, Yuanlin Guan, Kai Ye, and Gang Zhao

Corresponding Author(s): Kai Ye, Xi'an Jiaotong University

Review Timeline:

Submission Date:	July 8, 2021
Editorial Decision:	September 17, 2021
Revision Received:	October 25, 2021
Editorial Decision:	November 9, 2021
Revision Received:	November 10, 2021
Accepted:	November 10, 2021

Editor: Jan Claesen

Reviewer(s): Disclosure of reviewer identity is with reference to reviewer comments included in decision letter(s). The following individuals involved in review of your submission have agreed to reveal their identity: Edward Raby (Reviewer #2)

Transaction Report:

DOI: <https://doi.org/10.1128/Spectrum.00769-21>

September 17, 2021

Dr. Kai Ye
Xi'an Jiaotong University
Xi'an
China

Re: Spectrum00769-21 (Cerebrospinal fluids from healthy pregnant women does not harbor a detectable microbial community)

Dear Dr. Kai Ye:

Thanks for submitting your manuscript to Spectrum. Your work is definitely a good match for the journal and both reviewers generally liked the study, but have made some suggestions to improve the manuscript and further clarify outstanding questions. Please carefully address the reviewers' comments in a revised version and I would be happy to consider that.

Thank you for submitting your manuscript to Microbiology Spectrum. When submitting the revised version of your paper, please provide (1) point-by-point responses to the issues raised by the reviewers as file type "Response to Reviewers," not in your cover letter, and (2) a PDF file that indicates the changes from the original submission (by highlighting or underlining the changes) as file type "Marked Up Manuscript - For Review Only". Please use this link to submit your revised manuscript - we strongly recommend that you submit your paper within the next 60 days or reach out to me. Detailed information on submitting your revised paper are below.

Link Not Available

Sincerely,

Jan Claesen

Journals Department
Reviewer comments:

Reviewer #2 (Comments for the Author):

Thank you for the opportunity to review this comprehensive analysis of a well planned sample set. I have some suggestions to improve clarity of the manuscript.

Major comments

Reporting of redundant and non-redundant taxa and interpretation of their significance is unclear and inconsistent. For example lines 121-122 it is unclear if these are redundant/non-redundant taxa whereas elsewhere (e.g. lines 137-139) both are reported but the significance is unclear. Consider reporting sequentially in each section of the results or only reporting non-redundant routinely with a separate section providing a brief summary of redundant taxa, if deemed insignificant. Please also provide in methods the definitions used for redundant and non-redundant.

Although *Aspergillus* species are common opportunistic pathogens, *Aspergillus turcosus* has very rarely been reported from

clinical samples (see DOI: <https://doi.org/10.1128/MRA.01446-18>). Consider reframing discussion as the finding of this particular species may strengthen the argument of it being a contaminant.

Lines 280-281: please expand on method of random selection (computer generated sequence etc.?).

Lines 263-264: "providing a guide for disease diagnostics" do the authors suggest that a standardised method as used in this paper with sampling of saline and skin flora followed by subtraction is routinely employed in future clinical studies of CSF that use mNGS?

Lines 96-100 would be better placed in the discussion.

Lines 211-218 would be better placed in the discussion.

Minor comments

Figure 1 text: "replicates" is probably not correct here as I understand that each are individual samples rather than true replicates of the same sample.

Figure 2 text: consider expanding "other samples" to provide more complete description of the labels skin, swab, negative etc. as is done in the text for figure 1 so that this figure could stand alone.

Figure 3 text: again, consider expanding "negative" to include description (i.e. saline)

Although a reasonable label for analysis, "CSF_DNA" is unconventional in body text (e.g. line 121), consider replacing with "CSF-DNA" or "CSF DNA". Similarly some organism names should have "_" removed in body text e.g. "Escherichia_coli" (line 201).

Please carefully review spelling for example line 180 "substracted" -> "subtracted"; line 320 "palet" -> "pellet"; line 360 "equencing" -> "sequencing"; line 379 "organismal relative abundance" -> "relative abundance of organisms"; line 528 "inlet" -> "inset"; line 545 "remined" -> "remained"

Reviewer #3 (Comments for the Author):

Summary

"Cerebrospinal fluids from health pregnant women does not harbor a detectable microbial community" is a prospective study that aimed to describe the cerebrospinal fluid microbiome in healthy individuals. The study used CSF samples from 23 pregnant women aged 23-40 that were undergoing spinal anesthesia prior to C-section. Women were excluded from the study if they had other known infection or inflammatory condition, neurologic disease, hypertension, diabetes, heart disease or cancer. CSF was collected at the time of spinal anesthesia. Additionally skin and oral surface swabs were obtained to use as positive controls. A negative control of saline was collected in a syringe at the time of CSF collection.

Overall the study addresses an interesting question of whether or not a microbiome exists in the CSF. However, there are some study design issues that are not addressed in the manuscript and the manuscript would benefit greatly from review by an individual with a thorough knowledge of English grammar.

Major comments

1. While the manuscript addresses an important questions the study includes a very low number of study subjects and all of these participants are pregnant, adult, females. This small population significantly impairs the generalizability of the results of the study since only one gender is being examined in a narrow age group. Additionally, there are immunologic changes during pregnancy so that the growing fetus is tolerated by the mother. I fully understand how difficult it is to obtain CSF from normal health controls as this is an invasive procedure, however, the rationale for the convenience sample that was used in this study should be provided.
2. The manuscript would benefit from a more through discussion of the study design, this would help alleviate the above concern and bring overall clarity to the manuscript. It is unclear over what time period the study was conducted and how many women were considered but excluded based on the exclusion criteria. It would be helpful to discuss why the "n" of the study was 23. While it appears all 23 subjects had samples collected, only "twelve CSF samples were randomly selected for metastrascriptome studies (lines 280-281). It is extremely unclear if the final data set includes an "n" of 23 or 12.
3. The manuscript would benefit greatly from review by an individual with a thorough knowledge of English grammar. As it is written now the errors significantly impact the readability of the manuscript. Microbiome is frequently used in the incorrect context, cerebrospinal fluid should be used rather than cerebrospinal fluids, etc. There is inconsistent phrasing throughout the manuscript and the reference to positive, negative and contamination controls is not well delineated. At times CSF_DNA is used and this seems more like an input variable than an appropriate abbreviation for a manuscript. The methods, particularly sections on DNA extractions and purification, Metagenomics library construction and RNA library preparation for metatranscriptomics sequencing use a numbered approach that reads more like the steps in a protocol than a cohesive description of methods.
4. The manuscript would benefit from a more through discussion of the results, some speculation about the viruses and their

potential pathogenesis in humans would be beneficial. Since the only significant results in the CSF was that of *Aspergillus* DNA fragments, this deserves a more in-depth discussion as *Aspergillus* is a significant human pathogen.

Minor comments

1. The manuscript would also benefit from a more through discussion of the "contamination" controls. These were briefly mentioned but not well outlined in the study design.
2. A more thorough discussion of how the various types of controls were used to exclude many of the organisms found in the CSF and more significant referencing of this method would be helpful.
3. While surface swabs of the skin and mouth were used as a positive control in this study, there was no discussion of if serum/blood would be a more appropriate positive control. Serum/blood would in some ways be much more similar to CSF in terms of composition and it is also technically thought to be sterile.
4. Figure 1 a is very busy, it would be most helpful to have a figure illustrating the various samples and overall study design. Perhaps a flow chart would be more appropriate as a standalone figure rather than inclusion with 1b-d.
5. Overall the size of the text in the figures is difficult to read as it is often very small.

Staff Comments:

Preparing Revision Guidelines

Please return the manuscript within 60 days; if you cannot complete the modification within this time period, please contact me. If you do not wish to modify the manuscript and prefer to submit it to another journal, please notify me of your decision immediately so that the manuscript may be formally withdrawn from consideration by Microbiology Spectrum.

Summary

“Cerebrospinal fluids from health pregnant women does not harbor a detectable microbial community” is a prospective study that aimed to describe the cerebrospinal fluid microbiome in healthy individuals. The study used CSF samples from 23 pregnant women aged 23-40 that were undergoing spinal anesthesia prior to C-section. Women were excluded from the study if they had other known infection or inflammatory condition, neurologic disease, hypertension, diabetes, heart disease or cancer. CSF was collected at the time of spinal anesthesia. Additionally skin and oral surface swabs were obtained to use as positive controls. A negative control of saline was collected in a syringe at the time of CSF collection.

Overall the study addresses an interesting question of whether or not a microbiome exists in the CSF. However, there are some study design issues that are not addressed in the manuscript and the manuscript would benefit greatly from review by an individual with a thorough knowledge of English grammar.

Major comments

1. While the manuscript addresses an important questions the study includes a very low number of study subjects and all of these participants are pregnant, adult, females. This small population significantly impairs the generalizability of the results of the study since only one gender is being examined in a narrow age group. Additionally, there are immunologic changes during pregnancy so that the growing fetus is tolerated by the mother. I fully understand how difficult it is to obtain CSF from normal health controls as this is an invasive procedure, however, the rationale for the convenience sample that was used in this study should be provided.
2. The manuscript would benefit from a more through discussion of the study design, this would help alleviate the above concern and bring overall clarity to the manuscript. It is unclear over what time period the study was conducted and how many women were considered but excluded based on the exclusion criteria. It would be helpful to discuss why the “n” of the study was 23. While it appears all 23 subjects had samples collected, only “twelve CSF samples were randomly selected for metatranscriptome studies (lines 280-281). It is extremely unclear if the final data set includes an “n” of 23 or 12.
3. The manuscript would benefit greatly from review by an individual with a thorough knowledge of English grammar. As it is written now the errors significantly impact the readability of the manuscript. Microbiome is frequently used in the incorrect context, cerebrospinal fluid should be used rather than cerebrospinal fluids, etc. There is inconsistent phrasing throughout the manuscript and the reference to positive, negative and contamination controls is not well delineated. At times CSF_DNA is used and this seems more like an input variable than an appropriate abbreviation for a manuscript. The methods, particularly sections on DNA extractions and purification, Metagenomics library construction and RNA library preparation for metatranscriptomics sequencing use a numbered approach that reads more like the steps in a protocol than a cohesive description of methods.
4. The manuscript would benefit from a more through discussion of the results, some speculation about the viruses and their potential pathogenesis in humans would be beneficial. Since the only significant results in the CSF was that of *Aspergillus* DNA fragments, this deserves a more in-depth discussion as *Aspergillus* is a significant human pathogen.

Minor comments

1. The manuscript would also benefit from a more thorough discussion of the “contamination” controls. These were briefly mentioned but not well outlined in the study design.
2. A more thorough discussion of how the various types of controls were used to exclude many of the organisms found in the CSF and more significant referencing of this method would be helpful.
3. While surface swabs of the skin and mouth were used as a positive control in this study, there was no discussion of if serum/blood would be a more appropriate positive control. Serum/blood would in some ways be much more similar to CSF in terms of composition and it is also technically thought to be sterile.
4. Figure 1 a is very busy, it would be most helpful to have a figure illustrating the various samples and overall study design. Perhaps a flow chart would be more appropriate as a standalone figure rather than inclusion with 1b-d.
5. Overall the size of the text in the figures is difficult to read as it is often very small.

Thank you for all your valuable comments. We have provided a response letter addressing all the issues raised by the reviewers. For clarity, all reviewer comments or quoted contents are in italicized fonts. A point-to-point response to each comment is provided in normal fonts. References to revised manuscript contents are also provided where needed. Please notice that the Figure or Supplementary Figure/Table IDs in the response letter are the new IDs in the revised manuscript.

REVIEWER COMMENTS

Response to reviewer #2 comments:

Major comments

1. *Reporting of redundant and non-redundant taxa and interpretation of their significance is unclear and inconsistent. For example lines 121-122 it is unclear if these are redundant/non-redundant taxa whereas elsewhere (e.g. lines 137-139) both are reported but the significance is unclear. Consider reporting sequentially in each section of the results or only reporting non-redundant routinely with a separate section providing a brief summary of redundant taxa, if deemed insignificant. Please also provide in methods the definitions used for redundant and non-redundant.*

RESPONSE: Thanks for your advice. To clarify the results concisely, we carefully revised the number of microbial taxa mentioned in the results section using nonredundant microbial taxa (lines 104-110, pages 5-6; lines 120-122, page 6; lines 131-133, page 7). To illustrate the difference of numbers caused by microbial types, we added a comparison of the number of redundant and nonredundant microbes in Fig. S2 in the supplemental material, and affirmed their definitions in the methods section (lines 382-385, page 18) and Fig. S2, respectively. All revisions were marked in the manuscript.

Fig. S2. Comparison of the number of redundant and nonredundant microbes in different types of specimens.

2. *Although *Aspergillus* species are common opportunistic pathogens, *Aspergillus turcosus* has very rarely been reported from clinical samples (see DOI: <https://doi.org/10.1128/MRA.01446-18>). Consider reframing discussion as the finding of this particular species may strengthen the argument of it being a contaminant.*

RESPONSE: Considering the suggestion, we have reframed the discussion about *Aspergillus* species (lines 224-230, page 11).

3. *Lines 280-281: please expand on method of random selection (computer generated sequence etc.?).*

RESPONSE: Thanks for your suggestion. We have corrected the description of random selection (lines 291-296, page 14). To validate whether the microbiome, if any detected in CSF, was physiologically active, metatranscriptomic sequencing for 12 of the pregnant women CSF specimens were performed. When processing different batches of CSF specimens, we randomly select specimens for both metagenomic and metatranscriptomic sequencing, and the remaining samples just for metagenomic sequencing. A total of 12 CSF specimens were performed using metagenomic and metatranscriptomic sequencing.

4. *Lines 263-264: "providing a guide for disease diagnostics" do the authors suggest that a standardised method as used in this paper with sampling of saline and skin flora followed by subtraction is routinely employed in future clinical studies of CSF that use mNGS?*

RESPONSE: We have modified this sentence at Lines 264-267, page 13. We mean that such findings shall strengthen our understanding of the internal environment of CSF in healthy people, which have great implications to human health especially for neurological disorders and infections, and help the further disease diagnostics, prevention, and therapeutics in clinical settings.

5. *Lines 96-100 would be better placed in the discussion.*

RESPONSE: Considering the suggestion, we have moved these sentences to the discussion section and revised them carefully (lines 261-264, pages 12-13).

6. *Lines 211-218 would be better placed in the discussion.*

RESPONSE: Considering the suggestion, we have moved these sentences to the discussion section and reframed discussion as the finding of *Aspergillus* species (lines 224-230, page 11).

Minor comments

7. *Figure 1 text: "replicates" is probably not correct here as I understand that each are individual samples rather than true replicates of the same sample.*

RESPONSE: Thanks for your advice. The number represented a total of 6 extraction buffer specimens were collected, not the replicates. We have corrected this error of Fig. 1 text.

8. *Figure 2 text: consider expanding "other samples" to provide more complete description of the labels skin, swab, negative etc. as is done in the text for figure 1 so that this figure could stand alone.*

RESPONSE: According to the suggestion, we reorganized the original Fig. 2 and made the original Fig. 2b independent as Fig. 4. We have added the description of "other samples" of Fig. 4 text in revised manuscript.

9. *Figure 3 text: again, consider expanding "negative" to include description (i.e. saline).*

RESPONSE: Thanks for your advice. We have added the description of "negative" of new Fig. 5 text in revised manuscript.

10. Although a reasonable label for analysis, "CSF_DNA" is unconventional in body text (e.g. line 121), consider replacing with "CSF-DNA" or "CSF DNA". Similarly some organism names should have "_" removed in body text e.g. "Escherichia_coli" (line 201).

RESPONSE: Thanks for your advice. We have deleted the "_" of these words in the revised manuscript.

11. Please carefully review spelling for example line 180 "substracted" -> "subtracted"; line 320 "palet" -> "pellet"; line 360 "equencing" -> "sequencing"; line 379 "organismal relative abundance" -> "relative abundance of organisms"; line 528 "inlet" -> "inset"; line 545 "remined" -> "remained".

RESPONSE: We are very sorry for these spelling errors and have corrected them carefully. The manuscript has been revised by a native English speaker.

Response to reviewer #3 comments:

Major comments

1. While the manuscript addresses an important questions the study includes a very low number of study subjects and all of these participants are pregnant, adult, females. This small population significantly impairs the generalizability of the results of the study since only one gender is being examined in a narrow age group. Additionally, there are immunologic changes during pregnancy so that the growing fetus is tolerated by the mother. I fully understand how difficult it is to obtain CSF from normal health controls as this is an invasive procedure, however, the rationale for the convenience sample that was used in this study should be provided.

RESPONSE: We are sorry for the limitations of the collected samples in this study. CSF is difficult to obtain in healthy individuals. Only 23 pregnant females who underwent intraspinal anesthesia before the caesarean section were enrolled. Because CSF puncture is an invasive surgery and will cause pain and damage to the subjects. Usually, only patients with CNS infections will receive lumbar punctures. Lumbar puncture in healthy people is rare. We have added this description in our manuscript at lines 254-258, page 12.

2. The manuscript would benefit from a more through discussion of the study design, this would help alleviate the above concern and bring overall clarity to the manuscript. It is unclear over what time period the study was conducted and how many women were considered but excluded based on the exclusion criteria. It would be helpful to discuss why the "n" of the study was 23. While it appears all 23 subjects had samples collected, only "twelve CSF samples were randomly selected for metatranscriptome studies (lines 280-281). It is extremely unclear if the final data set includes an "n" of 23 or 12.

RESPONSE: Thanks for your advice. A more detailed description of the enrolled individuals, including the time period of sample collection, and the criteria were added in the Material and method section (lines 269-303, pages 13-14). According to the exclusion criteria, we only collected eligible pregnant women for this study, and the number of pregnant women excluded was not counted. We added a flow-chart in the supplemental material (Fig. S1) to illustrate the study design and sample composition. To validate whether the microbiome, if any detected in CSF, was

physiologically active, metatranscriptomic sequencing for 12 of the pregnant women CSF specimens were performed. When processing different batches of CSF specimens, we randomly select specimens for both metagenomic and metatranscriptomic sequencing, and the remaining samples just for metagenomic sequencing. A total of 12 CSF specimens were performed using metagenomic and metatranscriptomic sequencing, and 11 CSF specimens were only sequenced for the metagenomic.

3. The manuscript would benefit greatly from review by an individual with a thorough knowledge of English grammar. As it is written now the errors significantly impact the readability of the manuscript. Microbiome is frequently used in the incorrect context, cerebrospinal fluid should be used rather than cerebrospinal fluids, etc. There is inconsistent phrasing throughout the manuscript and the reference to positive, negative and contamination controls is not well delineated. At times CSF_DNA is used and this seems more like an input variable than an appropriate abbreviation for a manuscript. The methods, particularly sections on DNA extractions and purification, Metagenomics library construction and RNA library preparation for metatranscriptomics sequencing use a numbered approach that reads more like the steps in a protocol than a cohesive description of methods.

RESPONSE: Thanks for your advice. The manuscript has been revised by a native English speaker. Also, a more cohesive detailed description of the controls and methods has been added in the revised manuscript.

4. The manuscript would benefit from a more through discussion of the results, some speculation about the viruses and their potential pathogenesis in humans would be beneficial. Since the only significant results in the CSF was that of Aspergillus DNA fragments, this deserves a more in-depth discussion as Aspergillus is a significant human pathogen.

RESPONSE: Considering the suggestion, we have reframed the discussion about Aspergillus species (lines 224-230, page 11).

Minor comments

1. The manuscript would also benefit from a more through discussion of the "contamination" controls. These were briefly mentioned but not well outlined in the study design.

RESPONSE: Thanks for your suggestion. We clearly defined the contamination controls in the revised manuscript and added description in discussion section (Fig. S1; lines 99-104, page 5; lines 241-253, pages 12).

2. A more thorough discussion of how the various types of controls were used to exclude many of the organisms found in the CSF and more significant referencing of this method would be helpful.

RESPONSE: Considering the suggestion. We have reframed the discussion section. We describe the results and reasons of using three types of controls (negative controls: normal saline specimens; positive controls: skin specimens; contamination controls: DNA extraction buffer specimens) to remove the microbes detected in the CSF DNA specimens (lines 216-223, pages 10-11).

3. While surface swabs of the skin and mouth were used as a positive control in this study, there was no discussion of if serum/blood would be a more appropriate positive control. Serum/blood would in some ways be much more similar to CSF in terms of composition and it is also technically thought

to be sterile.

RESPONSE: Thanks for your constructive suggestion. The discussion of blood and the potential controls were added in the discussion section (lines 241-250, page 12).

4. Figure 1 a is very busy, it would be most helpful to have a figure illustrating the various samples and overall study design. Perhaps a flow chart would be more appropriate as a standalone figure rather than inclusion with 1b-d.

RESPONSE: According the suggestion, we have divided the original Fig. 1 into Fig. 1 and Fig. 2, and added a flow-chart (Fig. S1) in the supplemental material to illustrate the study design and sample composition.

Fig. S1. Flow-chart to illustrate the study design and sample composition. CSF and matched control specimens (positive controls: oral and skin; negative controls: saline solution) collected from 23 pregnant women along with sterile swabs and DNA/RNA extraction buffers (contamination controls) were sequenced for metagenomic and metatranscriptomic analysis.

5. Overall the size of the text in the figures is difficult to read as it is often very small.

RESPONSE: According the suggestion, we have enlarged the size of the text in the all advised figures.

November 9, 2021

Prof. Kai Ye
Xi'an Jiaotong University
Xi'an
China

Re: Spectrum00769-21R1 (Cerebrospinal fluid from healthy pregnant women does not harbor a detectable microbial community)

Dear Prof. Kai Ye:

Thanks for responding and addressing the reviewers' comments in this revised version of your paper. The Reviewer that evaluated your resubmission brought up two very minor points that you might want to address in a text-only revision before the paper gets accepted in its final form.

Thank you for submitting your manuscript to Microbiology Spectrum. As you will see your paper is very close to acceptance. Please modify the manuscript along the lines I have recommended. As these revisions are quite minor, I expect that you should be able to turn in the revised paper in less than 30 days, if not sooner. If your manuscript was reviewed, you will find the reviewers' comments below.

When submitting the revised version of your paper, please provide (1) point-by-point responses to the issues I raised in your cover letter, and (2) a PDF file that indicates the changes from the original submission (by highlighting or underlining the changes) as file type "Marked Up Manuscript - For Review Only". Please use this link to submit your revised manuscript. Detailed instructions on submitting your revised paper are below.

Link Not Available

Sincerely,

Jan Claesen

Reviewer comments:

Reviewer #2 (Comments for the Author):

Thank you for your revisions and response which has addressed all of the previous reviewers' comments.

Please check line 164 - is contamination suspected to have come from skin or from "skin samples". The current wording suggests contamination within the lab from skin samples.

Check spelling of Bray-Curtis line 548

Preparing Revision Guidelines

- point-by-point responses to the issues I raised in your cover letter
- Upload a compare copy of the manuscript (without figures) as a "Marked-Up Manuscript" file.
- Each figure must be uploaded as a separate file, and any multipanel figures must be assembled into one file.
- Manuscript: A .DOC version of the revised manuscript
- Figures: Editable, high-resolution, individual figure files are required at revision, TIFF or EPS files are preferred

Please return the manuscript within 60 days; if you cannot complete the modification within this time period, please contact me. If you do not wish to modify the manuscript and prefer to submit it to another journal, please notify me of your decision immediately so that the manuscript may be formally withdrawn from consideration by Microbiology Spectrum.

Thank you for your comments. We have provided a response letter addressing all the issues raised by the reviewer #2. Reviewer comments or quoted contents are in italicized fonts. A point-to-point response to each comment is provided in normal fonts.

REVIEWER COMMENTS

Response to reviewer #2 comments:

1. Please check line 164 - is contamination suspected to have come from skin or from "skin samples". The current wording suggests contamination within the lab from skin samples.

RESPONSE: Considering the suggestion, we have modified this sentence at line 164, page 8.

2. Check spelling of Bray-Curtis line 548.

RESPONSE: We are very sorry for this spelling error and have corrected “Bray-Cruits” at line 548, page 26.

November 10, 2021

Prof. Kai Ye
Xi'an Jiaotong University
Xi'an
China

Re: Spectrum00769-21R2 (Cerebrospinal fluid from healthy pregnant women does not harbor a detectable microbial community)

Dear Prof. Kai Ye:

Thanks for the quick response! I would like to congratulate you on the acceptance of your manuscript for publication in Spectrum.

Your manuscript has been accepted, and I am forwarding it to the ASM Journals Department for publication. You will be notified when your proofs are ready to be viewed.

Sincerely,

Jan Claesen
Editor, Microbiology Spectrum

Journals Department
Supplemental file 2: Accept
Supplemental file 1: Accept